# Transplantation of Brown Adipose Tissue with the Ability of Converting Omega-6 to Omega-3 Polyunsaturated Fatty Acids Counteracts High-Fat-Induced Metabolic Abnormalities in Mice

**DOI:** 10.3390/ijms23105321

**Published:** 2022-05-10

**Authors:** Tadataka Tsuji, Valerie Bussberg, Allison M. MacDonald, Niven R. Narain, Michael A. Kiebish, Yu-Hua Tseng

**Affiliations:** 1Section on Integrative Physiology and Metabolism, Joslin Diabetes Center, Harvard Medical School, Boston, MA 02215, USA; tadataka.tsuji@joslin.harvard.edu; 2BERG, Framingham, MA 01701, USA; valerie.bussberg@berghealth.com (V.B.); allison.macdonald@berghealth.com (A.M.M.); niven.narain@berghealth.com (N.R.N.); michael.kiebish@berghealth.com (M.A.K.); 3Harvard Stem Cell Institute, Harvard University, Cambridge, MA 02138, USA

**Keywords:** Fat-1, brown adipose tissue, transplantation, signaling lipidomics, ω-6 arachidonic acid-derived oxylipins

## Abstract

A balanced omega (ω)-6/ω-3 polyunsaturated fatty acids (PUFAs) ratio has been linked to metabolic health and the prevention of chronic diseases. Brown adipose tissue (BAT) specializes in energy expenditure and secretes signaling molecules that regulate metabolism via inter-organ crosstalk. Recent studies have uncovered that BAT produces different PUFA species and circulating oxylipin levels are correlated with BAT-mediated energy expenditure in mice and humans. However, the impact of BAT ω-6/ω-3 PUFAs on metabolic phenotype has not been fully elucidated. The Fat-1 transgenic mice can convert ω-6 to ω-3 PUFAs. Here, we demonstrated that mice receiving Fat-1 BAT transplants displayed better glucose tolerance and higher energy expenditure. Expression of genes involved in thermogenesis and nutrient utilization was increased in the endogenous BAT of mice receiving Fat-1 BAT, suggesting that the transplants may activate recipients’ BAT. Using targeted lipidomic analysis, we found that the levels of several ω-6 oxylipins were significantly reduced in the circulation of mice receiving Fat-1 BAT transplants than in mice with wild-type BAT transplants. The major altered oxylipins between the WT and Fat-1 BAT transplantation were ω-6 arachidonic acid-derived oxylipins via the lipoxygenase pathway. Taken together, these findings suggest an important role of BAT-derived oxylipins in combating obesity-related metabolic disorders.

## 1. Introduction

Obesity is a major risk factor for several metabolic diseases that represent the leading causes of disability and mortality, including type 2 diabetes [1,2], cardiovascular disease [3], non-alcoholic fatty liver disease [4] and cancers [5]. Dietary intake of fatty acids (FA) containing high levels of saturated FA and omega (ω)-6 polyunsaturated fatty acids (PUFAs) is one of the critical factors in developing obesity. PUFAs are the essential fatty acids for humans and other mammals because they are required for many biological processes but cannot be synthesized in the body and thus must be obtained from food sources [6]. Once ingested, PUFAs are presented as unesterified FA or esterified to complex lipids, such as phospholipids, cholesteryl esters, and triglyceride (TG). They can also be further metabolized by specific enzymes, such as cyclooxygenase (COX), lipoxygenase (LOX), and cytochrome P450 (CYP) epoxygenase, to form the oxygenated lipids, oxylipins [7].

Accumulated studies to date provide the fact that arachidonic acid (AA) derived from the ω-6 PUFAs promotes inflammation [8], while ω-3 PUFAs have beneficial effects on metabolic disorders, such as atherosclerosis, diabetes, and coronary heart disease by modulating cell proliferation and suppressing inflammatory response and oxidative stress [9,10]. In addition to the levels of these PUFAs in circulation, higher ω-6/ω-3 PUFA ratios are strongly associated with elevated pro-inflammatory states and increased risks of developing many cardiometabolic diseases [11,12,13]. Since mammals lack the enzyme converting ω-6 to ω-3 PUFAs, the Fat-1 transgenic mice were created by transgenic expression of the *C. elegans* Fat-1 gene, which encodes an ω-3 FA desaturase enzyme, in all tissues [14]. The Fat-1 transgenic mice can produce ω-3 PUFAs from the ω-6 FA source, leading to reduced ω-6/ω-3 ratios in tissues and circulation even when fed with a diet that is high in ω-6 and low in ω-3 PUFAs. Due to increasing the endogenous ω-3 PUFAs and reducing the ω-6/ω-3 PUFA ratios, the Fat-1 transgenic mice display improved metabolic phenotypes upon high-fat and high-sucrose feeding [15]. In order to identify the biomarker involved in the health benefits associated with a balanced ω-6/ω-3 ratio, Astarita et al. compared the lipid profiles of Fat-1 transgenic mouse and littermates wild-type (WT) mice fed with a high ω-6 PUFA corn oil diet for six months [16]. They found that the Fat-1 mice have marked alterations in the CYP pathway and minor alterations in the LOX and COX pathways, resulting in increased levels of ω-3 oxylipins and decreased ω-6 oxylipins. These findings reveal that the Fat-1 mice and their WT littermates generate distinct FA profiles in circulation and different tissues/organs [17], and the lipid profiles derived from the Fat-1 mice could be used as potential circulating biomarkers for a healthier metabolic status.

Brown adipose tissue (BAT) is a unique fat tissue that specializes in thermogenic energy expenditure. In recent years, BAT has been recognized as a secretory organ that produces many factors, such as hormones, chemokines, extracellular vesicles, and signaling lipid mediators (i.e., lipokines), to facilitate inter-organ crosstalk and systemically influence glucose and FA metabolism [18]. We and others have identified several altered oxylipins produced by BAT in response to physiological and pharmacological interventions, including cold exposure and administration of β3-agonist in humans and rodents [19,20]. BAT transplantation in mice provides a tractable gain-of-function approach to directly assess the impact of BAT on energy metabolism. Accumulating evidence suggests that successful BAT transplantation can reduce adiposity and improve glucose metabolism and insulin sensitivity in recipient mice [21,22,23,24,25,26]. In addition to whole fat tissue transplantation, delivery of brown or brown-like adipose precursor cells has also been demonstrated to benefit metabolism. Transplantation of CRISPR-engineered human brown-like (HUMBLE) preadipocytes can improve high-fat diet-induced metabolic syndrome in mice [27]. The metabolic improvements in the recipient are partly mediated via the release of various secreted factors, such as hormones, cytokines, and oxylipins, from the exogenous BAT or transplanted HUMBLE cells; however, the exact underlying mechanisms of the conveyed benefits have not been fully understood.

To directly determine the contributions of BAT-derived ω-3 PUFAs to systemic ω-6/ω-3 PUFAs ratios and metabolic phenotypes, we transplanted BAT from the Fat-1 transgenic mice and control WT animals into recipient mice fed with a high-fat diet and showed that the Fat-1 BAT-derived oxylipins may play a role in ameliorating high-fat diet-induced metabolic abnormalities.

## 2. Results

### 2.1. Fat-1 BAT Transplantation Increases Energy Expenditure and Improves High-Fat-Induced Glucose Intolerance

Activated BAT can produce multiple lipid mediators derived from ω-6 and ω-3 PUFAs [19,20]. To determine the metabolic impact of BAT-produced ω-3 PUFAs, we took advantage of Fat-1 transgenic mice, which can convert ω-6 PUFAs to ω-3 PUFAs, and transplanted BAT from the Fat-1 mice into recipient mice fed with a high-fat diet. We compared the metabolic phenotypes in the recipient mice transplanted with the Fat-1 BAT or WT BAT and the sham-operated mice (Figure 1). We utilized the Fat-1 transgenic mice and WT littermates as the donor for BAT transplantation. Those mice were fed with an ω-6-enriched diet containing 10% corn oil for 14 weeks. The recipients and sham-operated mice were 15-week-old C57BL/6J mice of the same gender, and they were put on a high-fat diet (HF diet, 60% kilocalories from lard) for 7 weeks after surgery to create an animal model with obesity-associated metabolic abnormalities.

All recipient male mice gained 20–30% of their original body weight during the 7 weeks of HF feeding (Figure 2A,B). Fat-1 BAT transplantation did not alter body weight. However, it significantly reduced the % of body weight gain in HF-fed male mice without changing food intake (Figure 2A–C). The changes in body weight were mainly caused by a reduction in tissue weight of perigonadal white adipose tissue (pgWAT). In contrast, tissue weight of subcutaneous WAT (scWAT), endogenous BAT, liver, and quadriceps muscle was not altered (Figure 2D). Notably, mice receiving Fat-1 BAT transplant had higher levels of oxygen consumption (VO_2_) and carbon dioxide production (VCO_2_) (Figure 2E–H) and greater heat production (Figure 2I,J) in both fed and fasted conditions, especially in the dark phase, compared to the sham-operated mice. Respiratory exchange ratio (RER) was not different among these groups (Figure 2K,L). HF feeding increases the levels of fasting glucose and insulin in circulation, leading to glucose intolerance and insulin resistance [28,29]. Mice carrying Fat-1 BAT transplants exhibited markedly improved glucose tolerance compared to the mice receiving WT BAT transplants or the sham group (Figure 2M,N). There were trends of lower circulating insulin levels, and Homeostatic Model Assessment for Insulin Resistance (HOME-IR) scores in the Fat-1 BAT transplanted group, but they did not reach statistical significance (Figure 2O,P).

To investigate whether the effects of Fat-1 BAT transplantation were sex specific, we performed the same surgical procedures and metabolic assessments as depicted in Figure 1 in female donors and recipient mice. Female Fat-1 BAT transplantation did not lead to significant changes in body weight gain and fat mass (Figure 3A–D). However, similar to what was observed in the male recipients, female mice receiving Fat-1 BAT transplants exhibited greater VO_2_ and VCO_2_ (Figure 3E–H) during HF feeding in the dark phase and significant improvement in glucose tolerance (Figure 3M,N) compared to the sham-operated and WT BAT-transplanted groups. By transplanting BAT of WT mice fed with an ω-6 PUFA-enriched diet for 14 weeks, the recipient mice showed higher fasting insulin levels and HOME-IR scores. Notably, the mice receiving Fat-1 BAT transplants displayed trends of lower insulin levels and HOME-IR scores than those carrying WT BAT transplants (Figure 3O,P). These results suggest that the metabolic benefits of Fat-1-BAT transplantation appeared to be more robust in male than female mice, although the effects on glucose tolerance, VO_2_, and VCO_2_ were equally efficacious in both sexes.

### 2.2. Fat-1 BAT Transplantation Increases the Expression of Genes Involved in Nutrient Uptake and Thermogenesis in Endogenous BAT of the Recipients

Next, to examine how Fat-1 BAT transplantation causes increases in energy metabolism without altering body weight and food intake, we measured the expression of genes involved in thermogenesis as well as lipid and glucose metabolism in the endogenous BAT of the recipient animals. Endogenous BAT of the mice receiving Fat-1 BAT transplants showed increases in the expression of genes related to the thermogenic phenotype, including uncoupling protein 1 (Ucp1), cell death-inducing DNA fragmentation factor, alpha subunit-like effector A (Cidea), PR domain-containing 16 (Prdm16), peroxisome proliferator-activated receptor gamma (Pparγ), and β-oxidative genes, including carnitine palmitoyltransferase 1b (Cpt1b), which is the rate-limiting enzyme for transporting FA into mitochondria, and peroxisome proliferator-activated receptor alpha (Pparα) (Figure 4A), consistent with the observed increases in energy expenditure. In addition, mRNA levels of vascular endothelial growth factor α (Vegfα), a vascularization marker, were significantly upregulated in endogenous BAT by receiving Fat-1 BAT transplants (Figure 4A).

As mitochondrial respiration and β-oxidation increase, the intracellular lipids need to be replenished by TG-derived FA through lipolysis and glucose uptake as well as FA intake, and then the fuels are utilized to produce heat in brown adipocytes. The mRNA levels of representative lipolytic genes, including adipose triglyceride lipase (Atgl) and hormone-sensitive lipase (Hsl), did not alter, while the expression of fatty acid transport protein 1 (Fatp1) and scavenger receptor Cd36 (which enhances cellular fatty acid uptake) and lipoprotein lipase (Lpl) (which converts TG to FA), as well as glucose transporter 1 (Glut1), was significantly upregulated in the endogenous BAT of mice receiving Fat-1 BAT transplants (Figure 4B). Among some selected de novo lipogenic genes, the expression of carbohydrate responsive-element-binding protein alpha (Chrebpα) mRNA was increased in the endogenous BAT of Fat-1 BAT recipient mice (Figure 4B). Intriguingly, the circulating TG levels were higher in mice receiving WT-BAT than that in sham-operated mice, while mice with Fat-1 BAT had similar levels of plasma TG levels as the sham-operated animals (Figure 4C). The levels of free FA in the endogenous BAT were higher in mice receiving Fat-1 BAT transplants than in mice carrying WT BAT transplants (Figure 4D), although no alterations of TG levels were found in endogenous BAT (Figure 4E).

Furthermore, we investigated the same selective genes involved in thermogenesis, browning, and lipid and glucose metabolism in the endogenous scWAT. As shown in Figure 4F,G, there were no statistical differences among the three groups. Hence, these data suggest that Fat-1 BAT transplantation activates the endogenous BAT, but not scWAT, by increasing its nutrient uptake and utilization abilities.

### 2.3. Altered Circulating Signaling Lipids Profiles in Recipients Carrying Fat-1 BAT Transplants

BAT secretes signaling molecules such as oxylipins from PUFAs that facilitate inter-organ crosstalk and regulate glucose and FA metabolism [18]. To determine the oxylipin secretomes produced by WT BAT and Fat-1 BAT, we profiled 113 signaling lipids in the plasma of the recipient mice using targeted liquid chromatography-tandem mass spectrometry (LC–MS/MS) lipidomic analysis.

Among the detectable signaling lipids, we found that the levels of six ω-6 PUFA-derived, and one ω-3 PUFA-derived oxylipins, were significantly elevated in mice receiving the WT BAT transplants compared with the sham-operated mice (Figure 5A,B). All these significantly altered oxylipins are metabolites of the LOX enzymes. These include the ω-6 AA-derived oxylipins 8-hydroxyeicosatetraenoic acid (HETE), 12-HETE, 15-HETE, 12-dehydro leukotriene B4 (12-oxo LBT4) and tetranor 12-HETE, the ω-6 dihomo-γ-linolenic acid (DGLA)-derived 15-HETrE (15-hydroxy-8Z,11Z,13E-eicosatrienoic acid), and the ω-3 EPA metabolite 12-hydroxyeicosapentaenoic acid (12-HEPE) (Figure 5C,D). Since the donor mice were fed with an ω-6-enriched diet, these data suggest that the lipids produced by the transplanted BAT were influenced by the type of diet given to the donors.

Next, we compared the plasma oxylipin profiles in mice receiving Fat-1 BAT vs. WT BAT. Nine of the 96 detected oxylipins were significantly decreased in the circulation of mice receiving Fat-1 BAT relative to mice receiving WT BAT transplants (Figure 6A,B). Of the altered ω-6 oxylipins, six AA and DGLA-derived oxylipins, 5-HETE, 12-HETE, 15-HETE, 12-oxo LTB4, PGK2, and 15-HETrE, were notably lower in mice receiving Fat-1 BAT transplants compared to the WT BAT transplanted mice (Figure 6B,C). Unexpectedly, we found that the levels of three ω-3 PUFA-derived oxylipins, 12-HEPE, 13-HDHA, and 14-HDHA, were lower in the circulation of mice carrying Fat-1 BAT than in mice receiving WT BAT transplants (Figure 6D), indicating a complex lipid dynamic occurring in the recipient animals.

## 3. Discussion

BAT has a significant impact on systemic metabolism; therefore, it becomes an attractive target to combat obesity and metabolic disorders. The present study reveals that transplantation of Fat-1 BAT, which is capable of converting ω-6 to ω-3 PUFAs, ameliorates high-fat-induced metabolic abnormalities. Presumably, this is mediated, at least in part, by increasing nutrient uptake and utilization abilities as well as the thermogenic capacity of the endogenous BAT in the recipients (Figure 7). These data also highlight a potential crosstalk between the transplanted BAT and endogenous brown fat.

Oxylipins are produced via enzymatic or non-enzymatic oxygenation of both ω-6 and ω-3 PUFAs. Three major enzymatic pathways are involved in their generation: COX, LOX, and CYP [7]. These pathways serve as critical pharmacological targets for various diseases. We have previously discovered that activated BAT can produce oxylipins to regulate lipid and glucose metabolism. Of these oxylipins, 12,13-dihydroxy-9Z-octadecenoic acid (12,13-diHOME), an ω-6 linoleic acid (LA)-derived oxylipin of CYP450, supports thermogenesis by shuttling FA into brown adipocytes for facilitating lipid utilization, and administration of 12,13-diHOME into obesity mice lowers circulating TG levels [19]. 12-HEPE, an ω-3 eicosapentaenoic acid (EPA)-derived oxylipin of LOX, promotes glucose uptake into BAT and skeletal muscle via activation of the PI3K-mTOR-Akt-Glut4 pathway, leading to improved glucose homeostasis in mice [20]. Thus, BAT appears to be one of the critical sites for producing oxylipins from PUFAs [30].

Fat-1 transgenic mice carry the *C. elegans* Fat-1 gene that encodes an ω-3 PUFAs desaturase to catalyze the conversion of ω-6 to ω-3 PUFAs [14]. The Fat-1 mice have been widely used to evaluate the beneficial effects of ω-3 PUFAs and the influence of a balanced ω-6/ω-3 PUFAs ratio on various physiological and pathophysiological states [31,32]. Numerous studies have shown that an imbalanced ω-6/ω-3 PUFAs ratio links to chronic inflammatory diseases and cancers [33]. The Fat-1 transgenic mice have a significant increase in ω-3 PUFA metabolites and marked decreases in several ω-6 phospholipids and TG levels in the plasma relative to WT mice [16]. Consistent with these findings, in this study, we discovered that mice receiving Fat-1 BAT transplants displayed decreased levels of ω-6 oxylipins compared with the mice receiving WT BAT transplants. Because the donor WT and Fat-1 mice were fed with an ω-6-enriched diet for 14 weeks, and Fat-1 BAT can convert ω-6 PUFAs to ω-3 PUFAs, the WT BAT and Fat-1 BAT represent two different donor tissues containing high and low levels of ω-6 PUFAs, respectively. Interestingly, compared with the sham-operated mice, the mice receiving WT BAT transplants had higher levels of ω-6 oxylipins in circulation, indicating that types of diet could modulate the secretomes of donor BAT. Although we cannot distinguish the lipids produced by the transplanted BAT from the endogenous sources in the current experimental settings, these results support the notion that BAT is a secretory organ and could contribute to circulating oxylipin profiles. The application of using stable isotope labeling of the donor BAT could help trace the metabolites produced by the transplants and warrants future investigation.

Although there are no significant differences in metabolic phenotype between the recipient mice carrying WT BAT transplants and the sham-operated mice, mice receiving Fat-1 BAT transplants exhibited considerable improvements in glucose tolerance, higher energy expenditure, and reduced plasma TG levels relative to the other two groups of mice upon high-fat feeding. Consistent with these metabolic phenotypes, expression of genes involved in thermogenesis and nutrient uptake and utilization was increased in the endogenous BAT of mice receiving Fat-1 BAT, suggesting that the transplants may activate recipients’ BAT to enhance energy metabolism. We previously reported that the CRISPR-engineered HUMBLE adipocytes could produce nitric oxide to activate endogenous BAT in recipient mice [27]. Thus, these findings illustrate dynamic interactions between transplanted tissues/cells and the recipient’s endogenous organs. Oxylipins exert positive or negative effects on inflammation, and nutrient metabolism [34,35,36]. Of the significantly altered oxylipins in the circulation of the recipient animals, we found that three ω-6 AA-induced oxylipins (12-oxo LBT4, 12-HETE, and 15-HETE) and one ω-6 DGLA-induced lipid (15-HETrE) were elevated in the plasma of the mice receiving WT BAT transplants relative to the sham-operation, and the levels of these oxylipins were decreased by Fat-1 BAT transplants compared with WT transplants. Exactly how these oxylipins are involved in the metabolic regulation of the recipient animals is currently unknown and is an important topic for future studies.

In conclusion, in this study, we demonstrated that the transplanted BAT significantly contributes to the recipient’s circulating lipid profiles, and diets used in feeding donor mice influence the signaling lipid profiles in the recipient animals. Transplantation of Fat-1 BAT reduces high-fat diet-induced metabolic disorders. Taken together, these findings highlight the critical role of BAT-derived oxylipins in the regulation of systemic energy metabolism and their potential effects on combating obesity-related metabolic diseases.

## 4. Materials and Methods

### 4.1. Animals, Surgical Procedures and Metabolic Assessments

All animal experiments and care procedures were approved by the Institutional Animal Care and Use Committee at Joslin Diabetes Center. C57BL/6-Tg(CAG-Fat-1)1Jxk/J (Stock no. 020097) and C57BL6/J (Stock no. 000664) mice were purchased from the Jackson Laboratory as breeder mice. By crossing them, heterozygous Fat-1 mice and littermates wild-type (WT) mice were obtained as donors. Mice were kept in a temperature-controlled room (23 °C) on a 12 h light/dark cycle (lights on 06:30 am; off 06:30 pm) with free access to food and water. The mice for donor were fed AIN-76A w/10% Corn Oil Diet (5T9W) (Scott Pharma, Inc), which has an enriched omega-6 content.

For BAT transplantation, 0.2 g BAT was removed from the interscapular region of donors of Fat-1 mice or WT mice (18 weeks old) after euthanasia. BAT was transplanted into the visceral cavity in each C57BL/6 recipient mouse under anesthesia. The transplant was carefully lodged deep within the perigonadal fat of the recipient following the procedures described in [26]. The sham operated mice received the same procedure, but instead of transplanting BAT, their perigonadal fat pad was exposed and then closed. The recipient C57BL6/J mice were fed a high-fat diet (60% kcal from fat, Research Diets #D12492) right after the transplantation or sham operation for 7 weeks. These recipient mice underwent the following procedures.

Following 3 weeks after transplantation or sham surgery, mice were monitored by the Comprehensive Laboratory Animal Monitoring System (CLAMS) while individually housed with ad libitum access to food and water for a period of 24 or 48 h to gather measurements of oxygen consumption rate (VO_2_), carbon dioxide production rate (VCO_2_), locomotor activity, respiratory exchange ratio (RER), food consumption, and energy expenditure (HEAT).

Following 6 weeks transplantation or sham surgery, an intraperitoneal glucose tolerance test (IPGTT) was performed. Mice were fasted for 6 h by transferring mice to clean cages without foods or feces in hoppers or bottom of cages. Mice had free access to drinking water. A baseline glucose level was determined by collecting blood from the tail of conscious mice before intraperitoneal glucose loading (2.0 g/kg body weight). Subsequently, blood was collected from the tail at 15, 30, 60, and 120 min after injection. Glucose concentrations were determined using a blood glucose meter (US Diagnostics).

Following 7 weeks after transplantation or sham surgery, blood and tissues were collected. Mice were fasted 6 h before sacrifice. Blood was obtained from each tail under anesthesia with inhalation of isoflurane (cat# NDC 66794-017-25, Piramal Critical Care) to determine fasting glucose concentrations by using a blood glucose meter (US Diagnostics). Blood was also collected by cardiac puncture and subsequently, plasma was separated by centrifugation at 4 °C and stored at −80 °C until future analysis of insulin and lipid mediators. The perigonadal and inguinal white fat, interscapular brown fat, liver, and quadriceps muscle were dissected and weighed, then snap-frozen in liquid nitrogen and stored at −80 °C until further analysis. The plasma insulin and triglyceride levels were determined by Ultra-Sensitive Mouse Insulin ELISA Kit (cat# 90080, Crystal Chem) and Triglyceride Assay kit (cat# ab65336, Abcam), respectively. Insulin resistance was assessed by calculating HOMA-IR ((fasting glucose × fasting insulin)/405). The levels of triglycerides and fatty acids in homogenized BAT were quantified by colorimetric methods (cat# ab65336 and ab65341, Abcam).

### 4.2. RNA Extraction and Quantitative PCR

Total RNA was extracted from tissues using Trizol reagent (Invitrogen), and RNA was purified using a spin column kit (cat# R2052, Zymo Research). RNA (500 ng-2 µg) was reverse-transcribed with a high-capacity complementary DNA (cDNA) reverse transcription kit (Applied Biosystems). Quantitative PCR (qPCR) was performed using SYBR Green PCR Master Mix (cat# A25778, Applied Biosystems) with 300 nM of each forward and reverse oligonucleotide primer in an ABI Prism 7900 sequence detection system (Applied Biosystems). Acidic ribosomal phosphoprotein P0 (Arbp) was selected as an internal standard in mice. Real-time PCR primer sequences are listed in Table 1.

### 4.3. Signaling Lipidomics

Plasma was analyzed by liquid chromatography-tandem mass spectrometry (LC–MS/MS) to semi-quantitatively measure the concentrations of a panel of 113 signaling lipids. Plasma samples were thawed at room temperature and immediately placed on ice. Aliquots of 100 µL were taken and added to 300 µL of methanol (stored at −20 °C) for a protein crash. A volume of 10 µL of a mixture of 5 deuterated internal standards, each at 100 pg/µL, was spiked into the samples, then samples were vortexed for 10 s and stored at −20 °C overnight. Samples were then subjected to solid-phase extraction. C18 cartridges at 500 mg/6 mL (Biotage, Uppsala, Sweden) were conditioned with 10 mL of methanol followed by 10 mL of water. The samples were centrifuged at 14,000× *g*, and the pH of the supernatant of the samples was adjusted by adding 3 mL of pH 3.5 water before loading the samples onto the C18 cartridges. The cartridges were washed with 5 mL water followed by 5 mL methylformate. Fractions were dried down under a stream of N2 gas and reconstituted in 50 µL methanol: water (1:1, by vol). Samples were vortexed and then transferred to LC–MS vials for analysis.

Electrospray ionization (ESI) LC–MS/MS was performed on a QTRAP 6500 (Sciex, Framingham, MA, USA) coupled to an Agilent Infinity 1290 (Agilent, Santa Clara, CA, USA) LC system with an InfinityLab Poroshell 120 EC-C18 (4.6 mm × 100 mm, 2.7 µm; Agilent) analytical LC column with a column oven heated to 60 °C. A volume of 10 µL of the sample was injected at a flow rate of 400 µL/min and were separated by reversed-phase chromatography with mobile phases A (100% H_2_O, 0.1% acetic acid) and B (100% MeOH, 0.1% acetic acid). The gradient started at 5% B and increased to 20% B by 3 min, then increased to 60% B by 8 min, increased to 90% B by 23 min, held at 90% B for 3 min, then returned to 5% B for the last 3 min. The total LC run time was 29 min. Samples were only acquired in negative polarity due to the chemical structure of the targeted lipids. The ESI source parameters were ion source gas 1 (GS1) 30, ion source gas 2 (GS2) 30, curtain gas (CUR) 30, ion spray voltage (IS) −4500, and temperature 500. The declustering potential (DP), entrance potential (EP), collision energy (CE), and exit potential (CXP) were tuned for each individually targeted lipid and internal standard. The MS method was a targeted scheduled MRM method. There were 101 MRM scans scheduled for optimized windows ranging from 120 to 360 s. The mass spectrometer acquisition time was set to 29 min. The LC–MS/MS data were acquired by Analyst 1.6.2 software (Sciex, Framingham, MA, USA) and processed with MultiQuant 3.0.1 (Sciex, Framingham, MA, USA) for peak integration. The targeted lipid species were measured semi-quantitatively and reported as area ratios of the peak area of the analyte divided by the peak area of the internal standard. Lipids that were detected in at least one sample were subjected for further analyses. By this definition, 96 out of the 113 signaling lipids were considered detectable.

### 4.4. Statistical Analysis

Data are shown as the mean values ± standard error of the mean (SEM). All statistics were performed by Prism 5.0 (GraphPad). Data were tested for a normal (Gaussian) distribution using Shapiro–Wilk normality test. Two comparisons were analyzed by using the two-tailed unpaired Student’s *t*-test. Multiple comparisons were performed by using one-way factorial ANOVA or two-way repeated-measures ANOVA followed by Tukey’s post hoc test. *p* values less than 0.05 were considered statistically significant.

## Figures and Tables

**Figure 1 ijms-23-05321-f001:**
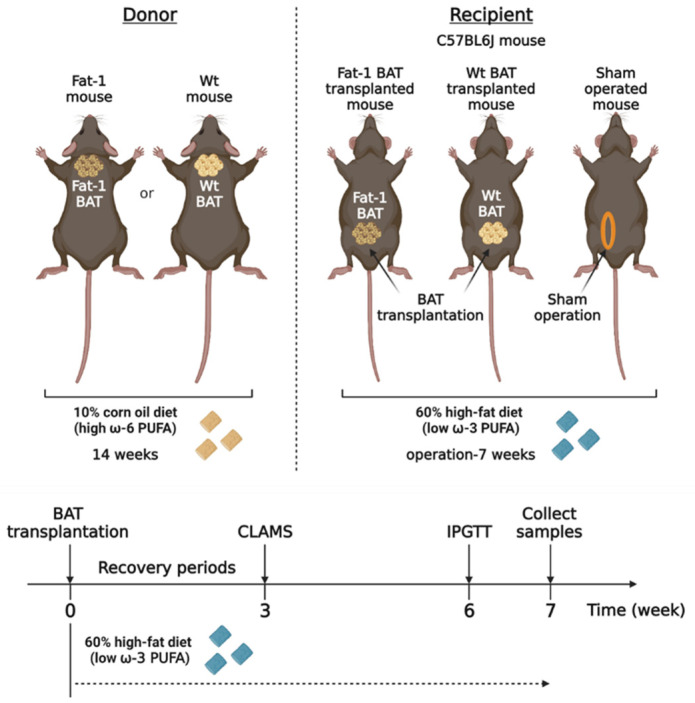
The experimental designs for brown adipose tissue (BAT) transplantation using the transgenic Fat-1 and control mice fed with an ω-6 PUFA-enriched diet. Schematic protocol for transplantation of BAT of the transgenic Fat-1 and wild-type (WT) mice fed with a diet containing 10% corn oil, which has an enriched ω-6 PUFA content, for a 14 week duration. All recipients and sham-operated mice were 15-week-old C57BL/6J mice on a high-fat diet (HF diet, 60% kilocalories from fat) for 7 weeks after surgical treatments. CLAMS: Comprehensive Laboratory Animal Monitoring System; IPGTT: intraperitoneal glucose tolerance test.

**Figure 2 ijms-23-05321-f002:**
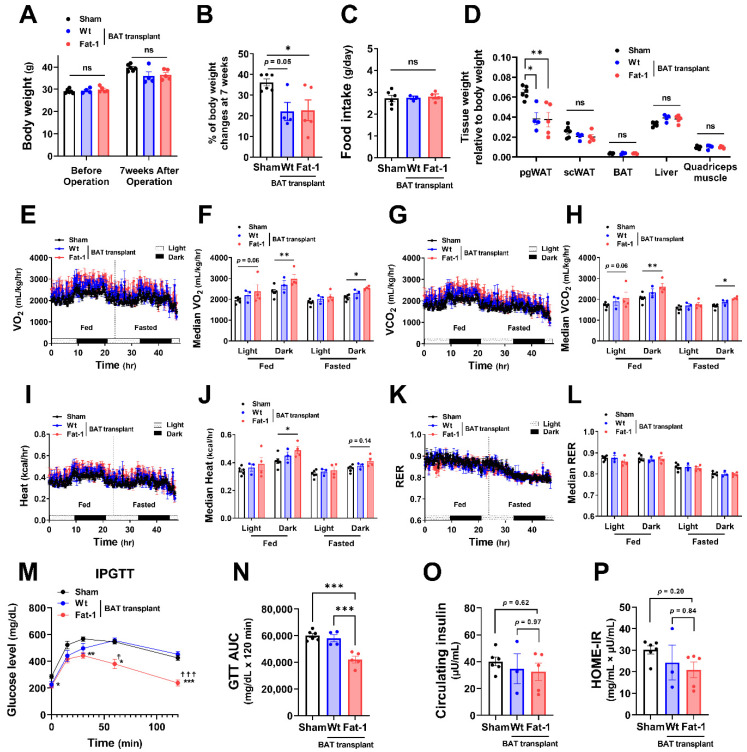
Metabolic impacts of Fat-1 BAT transplantation in male recipient mice. (**A**) Body weight of recipient mice before and 7 weeks after operation. (**B**) % of body weight changes 7 weeks after operation in mice receiving Fat-1 or wild-type (Wt) BAT transplant, or sham procedure. (**C**) Average daily food intake. (**D**) Tissue weight relative to body weight 7 weeks after operation. (**E**,**F**) Oxygen consumption (VO_2_), (**G**,**H**) carbon dioxide production (VCO_2_), (**I**,**J**) heat production, and (**K**,**L**) respiratory exchange ratio (RER) of mice receiving BAT transplantation or sham surgery 3 weeks after operation. (**M**) Glucose levels during intraperitoneal glucose tolerance test (IPGTT) 6 weeks after operation. * *p* < 0.05, ** *p* < 0.01, *** *p* < 0.001 Fat-1 BAT transplantation vs. sham-operation; ^☨^ *p* < 0.05, ^☨☨☨^ *p* < 0.001 Fat-1 BAT transplantation vs. WT BAT transplanted mice. (**N**) Area under the curve (AUC) of IPGTT. (**O**) Fasting plasma insulin and (**P**) homeostatic model assessment of insulin resistance (HOMA-IR) 7 weeks after operation. Data are represented as the mean ± SEM. Experiments were performed with *n* = 4–6 male mice per group. * *p* < 0.05, ** *p* < 0.01, *** *p* < 0.001, ns, not significant.

**Figure 3 ijms-23-05321-f003:**
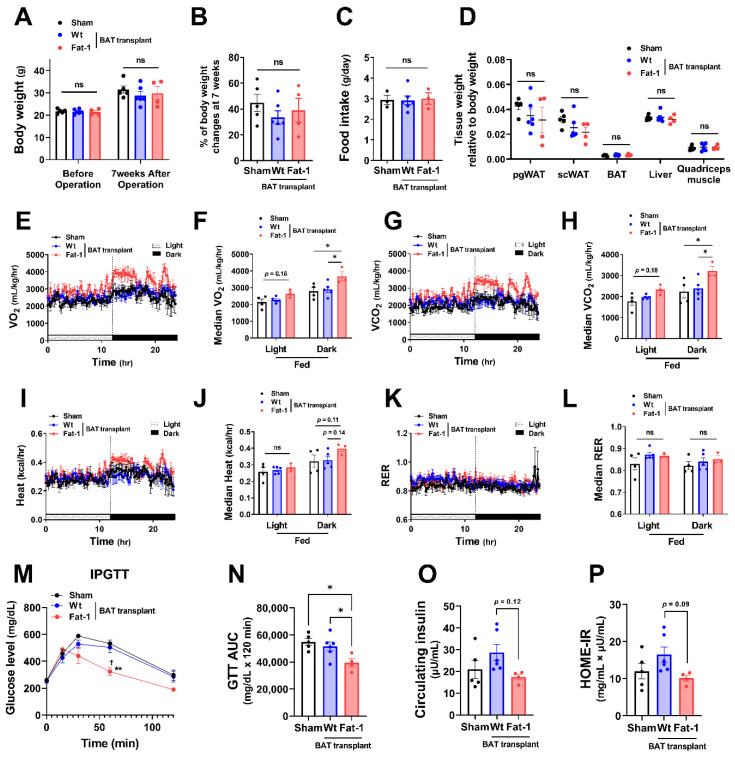
Metabolic impacts of Fat-1 BAT transplantation in female recipient mice. (**A**) Body weight of recipient mice before and 7 weeks after operation. (**B**) % of body weight changes 7 weeks after operation in mice receiving Fat-1 or wild-type (Wt) BAT transplant, or sham procedure. (**C**) Average daily food intake. (**D**) Tissue weight relative to body weight 7 weeks after operation. (**E**,**F**) Oxygen consumption (VO_2_), (**G**,**H**) carbon dioxide production (VCO_2_), (**I**,**J**) heat production, and (**K**,**L**) respiratory exchange ratio (RER) of mice receiving BAT transplantation or sham surgery 3 weeks after operation. (**M**) Glucose levels during intraperitoneal glucose tolerance test (IPGTT) 6 weeks after operation. ** *p* < 0.01 Fat-1 BAT transplantation vs. sham-operation; ^☨^ *p* < 0.05 Fat-1 BAT transplantation vs. WT BAT transplanted mice. (**N**) Area under the curve (AUC) of IPGTT. (**O**) Fasting plasma insulin and (**P**) homeostatic model assessment of insulin resistance (HOMA-IR) 7 weeks after operation. Data are represented as the mean ± SEM. Experiments were performed with *n* = 4–6 female mice per group. * *p* < 0.05, ns, not significant.

**Figure 4 ijms-23-05321-f004:**
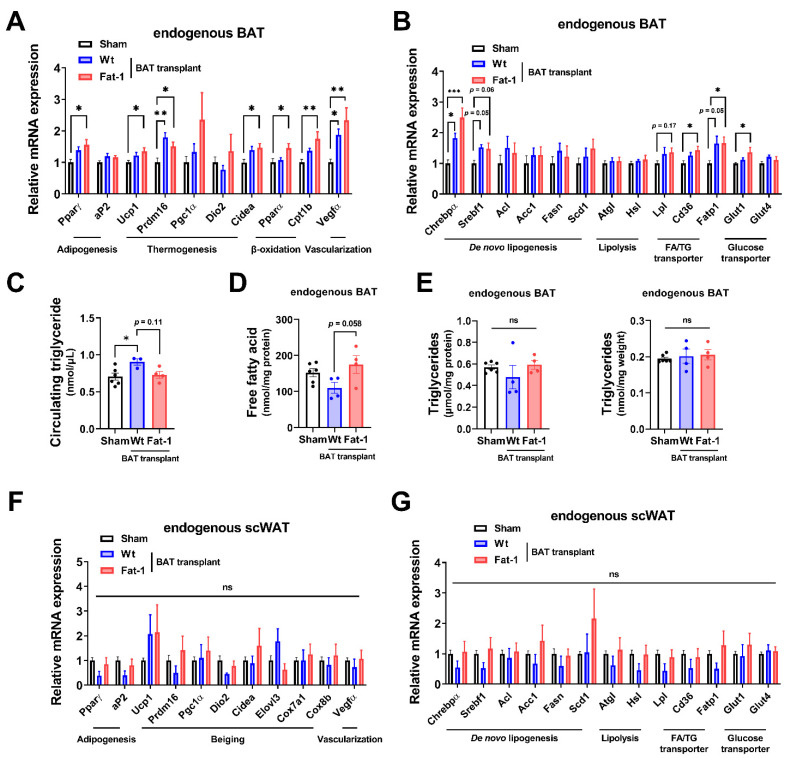
Fat-1 BAT transplantation leads to increased expression of genes involved in thermogenesis and fatty acid metabolism in endogenous BAT of the recipient mice. (**A**,**B**) Relative mRNA expression of indicated genes in endogenous BAT of male mice receiving BAT transplantation or sham surgery 7 weeks after operation. (**C**) Plasma triglyceride levels in recipient mice 7 weeks after operation. (**D**) Free fatty acid levels (**E**) and triglyceride levels (left: normalized to total protein; right: normalized to tissue weight) in endogenous BAT 7 weeks after operation. (**F**,**G**) Relative mRNA expression of indicated genes in scWAT of the same recipient mice 7 weeks after operation. Data are represented as the mean ± SEM. Experiments were performed with *n* = 4–6 mice per group. * *p* < 0.05, ** *p* < 0.01, *** *p* < 0.001, ns, not significant.

**Figure 5 ijms-23-05321-f005:**
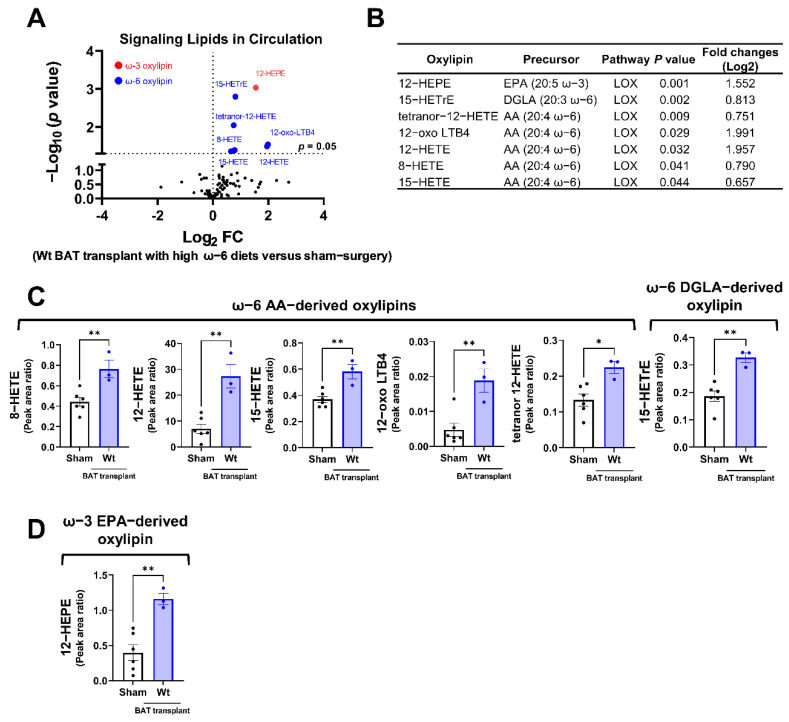
Targeted lipidomic analyses of circulating signaling lipids in mice receiving wild-type BAT transplants relative to the sham-operated mice. (**A**) Volcano plot of detectable signaling lipids profiled in male recipients transplanted with WT BAT from donors fed with a high ω-6 PUFA diet relative to sham-operated mice 7 weeks after operation. The dashed line indicates a *p* value of 0.05 (unpaired Student’s *t*-test). Significantly altered ω-3 and ω-6 oxylipins are highlighted in red and blue, respectively. (**B**) Characteristics of altered oxylipins, including precursor lipids and biosynthetic pathways. (**C**) Relative levels of ω-6 AA-derived oxylipins (8-HETE, 12-HETE, 15-HETE, 12-oxo-LTB4, and tetranor 12-HETE) and ω-6 DGLA-derived 15-HETrE. (**D**) Relative levels of ω-3 EPA-derived 12-HEPE. AA: arachidonic acid, DGLA: dihomo-γ-linolenic acid, EPA: eicosapentaenoic acid, and LOX: lipoxygenase. Data are represented as the mean ± SEM. Experiments were performed with *n* = 3–5 mice per group. * *p* < 0.05, ** *p* < 0.01.

**Figure 6 ijms-23-05321-f006:**
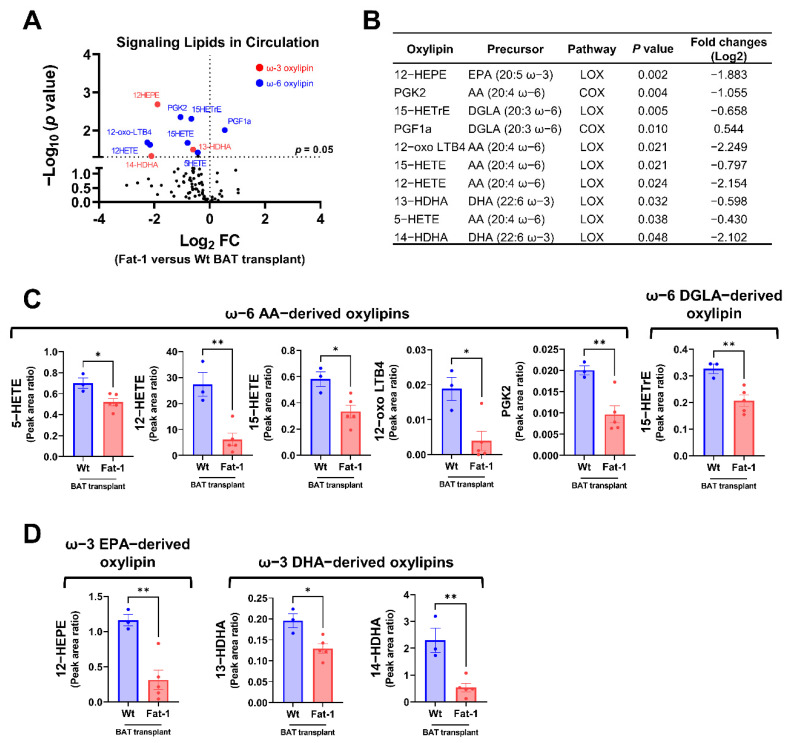
Targeted lipidomic analyses of circulating signaling lipids in mice receiving Fat-1 BAT transplants relative to mice receiving wild-type BAT transplants. (**A**) Volcano plot of detectable signaling lipids profiled in male recipients receiving Fat-1 transgenic BAT relative to mice receiving the WT BAT transplants 7 weeks after operation. The dashed line indicates a *p* value of 0.05 (unpaired Student’s *t*-test). Significantly altered ω-3 and ω-6 oxylipins are highlighted in red and blue, respectively. (**B**) Characteristics of altered oxylipins, including precursor lipids and biosynthetic pathways. (**C**) Relative levels of ω-6 AA-derived oxylipins (5-HETE, 12-HETE, 15-HETE, 12-oxo-LTB4, and PGK2), and ω-6 DGLA-derived 15-HETrE. (**D**) Relative levels of ω-3 EPA-derived 12-HEPE, and ω-3 DHA-derived oxylipins (13-HDHA and 14-HDHA). AA: arachidonic acid, COX: cyclooxygenase, DGLA: dihomo-γ-linolenic acid, DHA: docosahexaenoic acid, EPA: eicosapentaenoic acid, and LOX: lipoxygenase. Data are represented as the mean ± SEM. Experiments were performed with *n* = 3–5 mice per group. * *p* < 0.05, ** *p* < 0.01.

**Figure 7 ijms-23-05321-f007:**
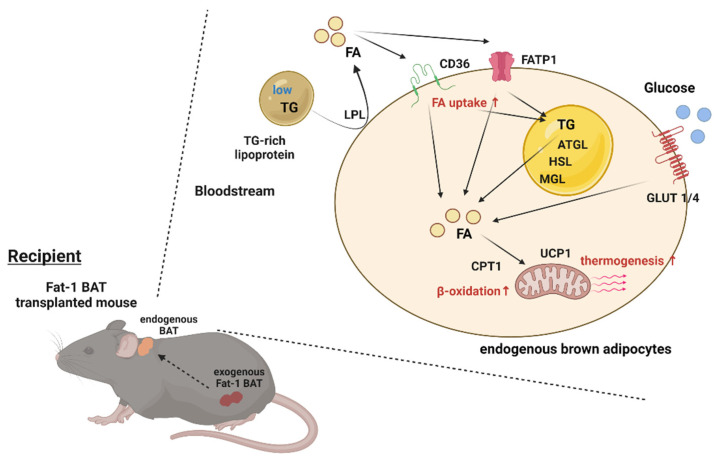
Proposed model of activation of the endogenous BAT via Fat-1 BAT transplantation. Schematic model showing proposed mechanism underlying activation of endogenous BAT of the recipient mice via Fat-1 BAT transplantation. ATGL: adipose triglyceride lipase, CPT1: carnitine palmitoyltransferase 1, FA: fatty acids, FATP1: fatty acid transport protein 1, GLUT1/4: glucose transporter 1/4, HSL: hormone-sensitive lipase, LPL: lipoprotein lipase, MGL: monoacylglycerol lipase, TG: triglyceride, and UCP1: uncoupling protein 1.

**Table 1 ijms-23-05321-t001:** Primer sequences.

Gene	Forward Primer	Reverse Primer
*Arbp*	TTTGGGCATCACCACGAAAA	GGACACCCTCCAGAAAGCGA
*Pparγ*	TCAGCTCTGTGGACCTCTCC	ACCCTTGCATCCTTCACAAG
*aP2*	AAGGTGAAGAGCATCATAACCCT	TCACGCCTTTCATAACACATTCC
*Ucp1*	CTGCCAGGACAGTACCCAAG	TCAGCTGTTCAAAGCACACA
*Prdm16*	CAGCACGGTGAAGCCATTC	GCGTGCATCCGCTTGTG
*Pgc1α*	CCCTGCCATTGTTAAGACC	TGCTGCTGTTCCTGTTTTC
*Dio2*	GCTGACCTCAGAAGGGCT	AGGTGGTCAGGTGGCTGA
*Cidea*	ATCACAACTGGCCTGGTTACG	TACTACCCGGTGTCCATTTCT
*Pparα*	GCGTACGGCAATGGCTTTAT	GAACGGCTTCCTCAGGTTCTT
*Cpt1b*	CCTGGTGCTCAAGTCATGGT	TGCTTGCACATTTGTGTTCTT
*Elovl3*	TCCGCGTTCTCATGTAGGTCT	GGACCTGATGCAACCCTATGA
*Cox7a1*	CAGCGTCATGGTCAGTCTGT	AGAAAACCGTGTGGCAGAGA
*Cox8b*	GAACCATGAAGCCAACGACT	GCGAAGTTCACAGTGGTTCC
*Vegfα*	GCTTCCTACAGCACAGCAGA	AATGCTTTCTCCGCTCTGAA
*Chrebpα*	CGACACTCACCCACCTCTTC	TTGTTCAGCCGGATCTTGTC
*Srebf1*	GCAGCCACCATCTAGCCTG	CAGCAGTGAGTCTGCCTTGAT
*Acl*	GCCAGCGGGAGCACATC	CTTTGCAGGTGCCACTTCATC
*Acc1*	CGGACCTTTGAAGATTTTGTCAGG	GCTTTATTCTGCTGGGTGAACTCTC
*Fasn*	GGCTCTATGGATTACCCAAGC	CCAGTGTTCGTTCCTCGG
*Scd1*	CCTGCGGATCTTCCTTATCA	GTCGGCGTGTGTTTCTGAG
*Atgl*	GTGAAGCAGGTGCCAACATTATTG	AAACACGAGTCAGGGAGATGCC
*Hsl*	CACCCATAGTCAAGAACCCCTTC	TCTACCACTTTCAGCGTCACCG
*Lpl*	GCCCAGCAACATTATCCAGT	GGTCAGACTTCCTGCTACGC
*Cd36*	ATGGGCTGTGATCGGAACTG	GTCTTCCCAATAAGCATGTCTCC
*Fatp1*	GTGCGACAGATTGGCGAGTT	GCGTGAGGATACGGCTGTTG
*Glut1*	CAGTTCGGCTATAACACTGGTG	GCCCCCGACAGAGAAGATG
*Glut4*	GTGACTGGAACACTGGTCCTA	CCAGCCACGTTGCATTGTAG

## Data Availability

All datasets generated for this study are included in this article.

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
