# Peer review of "Transplantation of Brown Adipose Tissue with the Ability of Converting Omega-6 to Omega-3 Polyunsaturated Fatty Acids Counteracts High-Fat-Induced Metabolic Abnormalities in Mice"

_ijms, 2022, doi:10.3390/ijms23105321_

Round 1

Reviewer 1 Report

The authors conducted a study on brown adipose tissue, which can convert omega-6 to omega-3 polyunsaturated fatty acids.
In vivo assays for gene and omega activity were performed using mice. The results obtained by comparing them with the results of many papers look interesting. It is judged that there are no modifications.

Author Response

Comments and Suggestions for Authors:

The authors conducted a study on brown adipose tissue, which can convert omega-6 to omega-3 polyunsaturated fatty acids. In vivo assays for gene and omega activity were performed using mice. The results obtained by comparing them with the results of many papers look interesting. It is judged that there are no modifications.

Response:

We graciously appreciate this reviewer for the positive feedback.

Reviewer 2 Report

This manuscript describes extensive amount of in vivo data about how transplantation of brown adipose tissue of Fat-1 mice affects the metabolism of polyunsaturated fatty acids in the host mice. Furthermore, the transplantation results in several beneficial metabolic effects in the recipients via the activation of their endogenous brown adipose tissue (but not beige adipocytes). The Authors performed elegant in vivo experiments and lipidomics analysis, the presented data is clinically relevant, and the manuscript is well written. I have only a few minor comments.

Specific suggestions:

  1. It should be included in the title that the observed phenomenon was found in mice.
  2. L41: Abbreviation of TAG should be defined here instead of lane 171.
  3. L82: “cytokines” instead of “cytokine”.
  4. L105: Abbreviation of BAT should be defined here instead of lane 107.
  5. All abbreviations found in Figure 7 should be defined in the corresponding legend.
  6. 4.4. How normality was assed? Please, add this information.

Author Response

Comments and Suggestions for Authors:

This manuscript describes extensive amount of in vivo data about how transplantation of brown adipose tissue of Fat-1 mice affects the metabolism of polyunsaturated fatty acids in the host mice. Furthermore, the transplantation results in several beneficial metabolic effects in the recipients via the activation of their endogenous brown adipose tissue (but not beige adipocytes). The Authors performed elegant in vivo experiments and lipidomics analysis, the presented data is clinically relevant, and the manuscript is well written. I have only a few minor comments.

Response: We appreciate the constructive input that our original manuscript received. In this revised manuscript, we have thoroughly proofread the manuscripts and addressed the reviewer’s comments and remarks. Please find below the point-by-point responses to this reviewer’s concerns.

Specific suggestions

1) It should be included in the title that the observed phenomenon was found in mice.

Response: We have changed the title. Here is the new title “Transplantation of brown adipose tissue with the ability of converting omega-6 to omega-3 polyunsaturated fatty acids counteracts high-fat induced-metabolic abnormalities in mice”.

2) L41: Abbreviation of TAG should be defined here instead of lane 171.

Response: The abbreviation for TAG has been defined.

3) L82: “cytokines” instead of “cytokine”.

Response: This typo has been fixed.

4) L105: Abbreviation of BAT should be defined here instead of lane 107.

Response: The abbreviation for BAT has been defined.

5) All abbreviations found in Figure 7 should be defined in the corresponding legend.

Response: All the abbreviations in Figure 7 have been defined in the figure legends.

6) 4.4. How normality was assed? Please, add this information.

Response: Data were tested for a normal (Gaussian) distribution using the Shapiro-Wilk normality test. We have added this description in section 4.4. Statistical analysis.